# CAMDLES: CFD-DEM Simulation of Microbial Communities in Spaceflight and Artificial Microgravity

**DOI:** 10.3390/life12050660

**Published:** 2022-04-29

**Authors:** Rocky An, Jessica Audrey Lee

**Affiliations:** 1Sibley School of Mechanical and Aerospace Engineering, Cornell University, Ithaca, NY 14850, USA; 2Department of Biological and Environmental Engineering, Cornell University, Ithaca, NY 14850, USA; 3Space Biosciences Division, NASA Ames Research Center, Moffett Field, CA 94035, USA

**Keywords:** microgravity, clinostat, rotating wall vessel, microbial community, population dynamics, computational fluid dynamics, CFD-DEM

## Abstract

We present CAMDLES (CFD-DEM Artificial Microgravity Developments for Living Ecosystem Simulation), an extension of CFDEM^®^Coupling to model biological flows, growth, and mass transfer in artificial microgravity devices. For microbes that accompany humans into space, microgravity-induced alterations in the fluid environment are likely to be a major factor in the microbial experience of spaceflight. Computational modeling is needed to investigate how well ground-based microgravity simulation methods replicate that experience. CAMDLES incorporates agent-based modeling to study inter-species metabolite transport within microbial communities in rotating wall vessel bioreactors (RWVs). Preexisting CFD modeling of RWVs has not yet incorporated growth; CAMDLES employs the simultaneous modeling of biological, chemical, and mechanical processes in a micro-scale rotating reference frame environment. Simulation mass transfer calculations were correlated with Monod dynamic parameters to predict relative growth rates between artificial microgravity, spaceflight microgravity, and 1 g conditions. By simulating a microbial model community of metabolically cooperative strains of *Escherichia coli* and *Salmonella enterica,* we found that the greatest difference between microgravity and an RWV or 1 g gravity was when species colocalized in dense aggregates. We also investigated the influence of other features of the system on growth, such as spatial distribution, product yields, and diffusivity. Our simulation provides a basis for future laboratory experiments using this community for investigation in artificial microgravity and spaceflight microgravity. More broadly, our development of these models creates a framework for novel hypothesis generation and design of biological experiments with RWVs, coupling the effects of RWV size, rotation rate, and mass transport directly to bacterial growth in microbial communities.

## 1. Introduction

### 1.1. Microbial Communities in Space

Microbial communities are an integral part of space exploration: they accompany human crew in habitats such as the International Space Station [1] and they will ultimately be used for bioregenerative life support [2], food and pharmaceuticals production [3], and in situ resource utilization [4]. Like all organisms, microbes experience stress in the space environment: studies have documented changes in virulence, antimicrobial resistance, and population diversity [5,6]. However, the mechanisms by which spaceflight induces these responses remain poorly understood. Many of the processes documented for multicellular organisms, such as mitochondrial dysregulation and mechanical unloading [7], do not apply to single-celled organisms; moreover, microorganisms almost invariably live in communities, but few studies of any organisms at any scale have investigated the effects of space conditions on interspecies interactions. Understanding the microbial space response requires an approach that considers the interactions between microbial cells and their environments at relevant spatial scales.

### 1.2. Validating Clinostats for Accurate Artificial Microgravity Conditions

Of the stressors present in the spaceflight environment, altered gravity is the feature most likely to affect microorganisms in habitable environments, such as a spacecraft. It is also the feature most difficult to replicate in ground studies. Since spaceflight experiments are costly, logistically difficult, and inaccessible to many researchers, artificial microgravity devices, such as clinostats and rotating wall vessel (RWV) bioreactors, are commonly used to replicate microgravity conditions. An RWV is a type of clinostat that is used for cell culture: they are cylindrical culture vessels filled with liquid culture medium that constantly rotate the fluid environment, keeping cells in suspension by continuously changing the orientation of the gravity vector [8,9]. However, there is a fundamental difference between microbial cells and larger organisms in the way that rotation is used to simulate microgravity. For plants grown in clinostats, the constant rotation of the gravity vector results in an average gravity vector of zero over time. For microbial cells, which are believed to be unable to sense gravity directly, it is the ability of the RWV to mimic the unique fluid environment of microgravity that is likely its most important feature [10,11,12,13,14,15]. Understanding the fluid dynamics of the RWV is therefore critical for conducting rigorous research into the effects of spaceflight on microorganisms.

### 1.3. The Microgravity Fluid Environment

The loss of natural convection in microgravity causes the fluid environment to be quiescent. Likewise, the idealized RWV fluid environment is under solid-body rotation, with minimal fluid shear and turbulence [8]. However, a disparity is observed in the cellular response when comparing RWV and microgravity conditions, which may be caused by a complex set of factors, including hardware effects, mechanotransduction, and nutrient stress [16,17]. RWVs also exhibit centrifugal sedimentation of cell aggregates, resulting in localization that is different from that found in true microgravity [8]. There are very few commercially available bioreactors; therefore, they are often custom-designed by researchers and unique to each researcher’s experiment. A general lack of knowledge about accurate RWV conditions is a problem that may be one of the sources of contradictory conclusions about the effect of microgravity among cell biological circles, assuming there even is a unified microgravity response on bacterial cells at all [18,19].

### 1.4. The Effect of Microgravity on Microbial Ecology

The interaction between the phases—that is, how microbial cells alter their fluid environment and vice versa—is especially important for understanding the behavior of multispecies microbial communities in space. No microbial species lives in isolation. Microorganisms form communities in which different species interact in numerous ways, and one of the interactions most commonly documented is the exchange of metabolites used for growth, often called cross-feeding [6,20]. In a quiescent microgravity fluid environment, reduced mixing could lead to a reduced exchange of metabolites, which, in turn, would shift community dynamics. Although a few microbial community experiments have been done in artificial microgravity [21], there are no previous studies that attempt to validate computational models for a quantitative understanding of microbial community dynamics in these devices. Numerous computational fluid dynamics studies on the macro-scale have been done to validate the ability of clinostats and RWV’s to produce the desired quiescent fluidic environment, but the scope of these studies often includes the fluid phase only, or solid parcel fractions, without metabolite scalar transport [22,23,24,25,26,27]. We sought to fill this gap by building a computational model that integrates both fluid and particle dynamics with biological metabolism and growth to generate predictions of microbial community dynamics in 1 g, microgravity, and RWV conditions. As an initial demonstration, we modeled a community composed of two obligately cross-feeding bacterial species that were previously studied and modeled extensively [20,28,29,30].

### 1.5. Agent-Based Models for Microbial Communities

Agent-based models, also known as individual-based models, operate at the resolution of the individual cell. Each agent, i.e., bacterial cell, is given certain rules to follow, placed in an environment, and expected to interact with that environment as well as with other agents. Complex interactions between agents, distributions, and patterns arise from these models, and they are essential for the modeling of microbial communities. Computational agent-based models in the microbial ecology field are actively being developed; one of the latest scientific agent-based simulations is NUFEB [31], which features biological growth kinetics and coupled fluid–solid interaction, but not solute scalar transport.

To apply the principles of agent-based modeling to spaceflight and artificial microgravity, we present CAMDLES (CFD-DEM Artificial Microgravity Developments for Living Ecosystem Simulation), which is an additional package for established, non-biological CFD-DEM software. It is based on the open-source software CFDEM^®^Coupling [32] and LIGGGHTS^®^ [33], and it brings CFD-DEM modeling capability to RWVs. CAMDLES features include a rotating reference frame simulation environment, the buoyant Boussinesq approximation for natural convective flows, and Monod kinetic metabolic models for calculation of biological growth rates. CAMDLES is also the first agent-based model that directly couples metabolism to mass transfer.

## 2. Materials and Methods

### 2.1. Software Development

CAMDLES is dependent on parent software packages to run simulations. Installation involves the addition of the package into the source code of the packages LIGGGHTS^®^ and CFDEM^®^Coupling. CAMDLES-specific scripts are provided in Appendix A. Table 1 and Figure 1 describe the parent software and relevant software packages. The parent packages are all written in the C++ programming language. Related software was used as guidance for CAMDLES development. All described software is open source and is distributed under the terms of the GNU Public License.

### 2.2. An Engineered Cross-Feeding Microbial Community

We simulated a previously described (Figure 2) engineered model microbial community involving two bacterial strains, *Escherichia coli* K12 Δ*metB* and *Salmonella enterica* ser. Typhimurium, that were extensively studied in 1 g but not in space conditions [20,28,29,30,40,41,42,43]. The *E. coli* strain is auxotrophic for the amino acid methionine and secretes acetate as a secondary metabolic product. The *S. enterica* depends on the secreted acetate for growth, and the strain was experimentally evolved to secrete methionine to support the *E. coli*. The coculture is therefore dependent upon cross-feeding when grown in a minimal medium containing lactose as the sole carbon source for growth.

Because of the organisms’ dependence on the flux of exchanged metabolites, the population dynamics of this model microbial community were found to depend on the spatial structure of the community [20]. For instance, colonies growing on agar medium achieve smaller sizes when located further away from each other, and a third colony blocking the exchange of metabolites between an *E. coli* colony and an *S. enterica* colony can cause a complex “metabolic eclipse” effect [28].

We hypothesized that in cross-feeding cocultures, reduced mixing in a quiescent microgravity fluid environment will enable high growth rates when the two species are colocalized in cooperative colonies, but only low growth rates when the species are located distantly from one another. We further hypothesized that in an RWV that simulates microgravity imperfectly, a variety of mechanical and fluid interactions will reduce the impact of colony localization on the growth rate.

### 2.3. Simulation Domain

In our simulations, we designed cooperative colonies with varying degrees of population density and localization. Limits in computing power necessarily made simulation timespans much shorter than the generation time of the species; we, therefore, considered the growth rate in terms of the cell mass increase rather than replication, in addition to metabolite production and consumption rates, to assess whether relative growth rates and mass transfer rates were sustainable in our prescribed cocultures. Additionally, CAMDLES does not support changes in diffusion based on solutes or solids, such as an extracellular matrix (ECM); therefore, to simulate the effect of such features, we were constrained to choosing a constant diffusion rate for each solute. We compared three diffusion rates: low diffusion seen in solid biofilms/ECM, intermediate diffusion, and high diffusion seen in liquid media (Appendix A). When the distinction is not stated, we set all solute diffusivities to an intermediate 5 × 10^−6^ cm^2^/s.

We chose our computational domain to be a box that was 300 μm × 100 μm × 300 μm in size, which is the smallest box that can host two cells at an inoculation cell density of 10^5^ cells/mL per species. A fluid parcel was 5 μm × 20 μm × 5 μm with a volume equal to that of ~430 bacterial cells when randomly packed. For some case studies, we varied the cell densities between 10^5^ and 10^8^ cells/mL, exploring a range of microbial population densities commonly achieved with this system in laboratory culture. During simulation initialization, cells were placed in colony-like formations: either a dense spherical mass of cells, or a mass with Gaussian random distributions around a sphere. We also tested random cell placements distributed throughout the computational domain, as well as a cylindrical colony formation as an approximation of a biofilm shape in the case of 1 g gravity. Periodic boundary conditions were set for the non-radial boundaries. In the radial direction, the outer wall was set as an inlet with zero velocity, and the inner wall was a freestream outlet. Cell metabolism was kickstarted by giving each species cell equivalents of product for the other species, i.e., *E. coli* was given acetate, and *S. enterica* was given methionine. This is similar to previous studies with this system using modeling with ordinary differential equations and is necessary to enable growth at initial timepoints [29]. In simulations with higher initial cell populations, we gave less starting product, ranging from 10^−2^ cell equivalents during inoculation, to 10^−8^ at maximum population. After initialization, each case study ran for two to five minutes, assuming cell growth to be exponential during this time. 

### 2.4. Mechanical Models

We assumed the volume and density of a bacterial cell to be the same for both species, namely, *E. coli* and *S. enterica*. The volume was calculated as if the cell was a spherocylinder, but was converted to an equivalent Stokes radius (1) and approximated as a sphere in silico.
(1)$$r={\left(\frac{3V_{cell}}{4\pi }\right)}^{\frac{1}{3}}$$

The Stokes drag for a very low Reynolds number was applied to each bacterial cell (2). $\vec{v_{s}}$ and $\vec{u_{f}}$ are the solid and fluid velocities, respectively. In this study, $u_{f}$ in (2) was assumed to be negligible. Because of the low inertia, the solid–fluid momentum coupling caused by the Stokes drag was also assumed to be negligible.
(2)$$F_{viscous}=6\pi r\mu \left(v_{s}-u_{f}\right)$$

In the case of artificial microgravity, the simulation box was placed in a rotating reference frame that matched the rotation rate of the RWV (see Figure 3). Solid-body fluid rotation was enforced at the beginning of the artificial microgravity simulation. The net body frame acceleration ${\vec{a}}^{B}$ was derived (3) using centrifugal, Coriolis, gravity, and drag accelerations for the cells in the rotating frame. Archimedean buoyancy was also accounted for in the centrifugal and gravity terms by use of specific gravity. The last term is the viscous drag, with $\frac{\rho_{f}}{\rho_{s}}$ being the fluid–solid density ratio and ν being the kinematic liquid viscosity.
(3)$${\vec{a}}^{B}=-\vec{\omega }\times \left(\vec{\omega }\times \vec{r^{B}}\;\right)\left(1-\frac{\rho_{f}}{\rho_{s}}\right)-2\left(\vec{\omega }\times \vec{v^{B}}\right)+\vec{g}\left(1-\frac{\rho_{f}}{\rho_{s}}\right)-\frac{9}{2}\frac{\rho_{f}}{\rho_{s}}\nu \left(\vec{v_{s}}-\vec{u_{f}}\right)\frac{1}{r^{2}}$$

Because $\vec{r^{B}}$ spans the range of only ±1.5%, $\vec{r^{B}}$ was set to a constant extending to the center of the simulation box. The centrifugal acceleration term was approximated as a uniform body force $\vec{a_{c}}$. In this study, we set the radius of rotation to be 1 cm, positioning the computational domain within an imagined culture vessel with a rotation rate of 10 R.P.M. (common for bacteria in artificial microgravity experiments).

The rotating reference frame acceleration was also applied to the fluid as a modification of the left-hand side of the incompressible Navier–Stokes Equation (4). The Boussinesq approximation for natural, solutal, and convective flows was applied to the right-hand side of the equation; this assumed that solutal changes in fluid density were small enough to affect buoyancy force only and neither the inertia nor viscosity [40]. For this study, this approximation was included three times, once for each metabolite in the media: acetate, methionine, and lactose. ε is the local fluid volume fraction.
(4)$$\begin{array}{cc} \frac{\partial \varepsilon \vec{u_{f}}}{\partial t}+\left(\varepsilon \vec{u_{f}}\cdot \;\nabla \right)\vec{u_{f}} & +\;\varepsilon \left[\vec{\omega }\times \left(\vec{\omega }\times \vec{r^{B}}\;\right)+2\left(\vec{\omega }\times \vec{v^{B}}\right)\right]\\ = & -\varepsilon \nabla \frac{p_{rgh}}{\rho }+\nu \nabla \cdot \left(\nabla \varepsilon \;\vec{u_{f}}\right)+\varepsilon \left(\vec{g}+\vec{a_{c}}\right)\left[\beta_{Si}\left(S_{i}-S_{iR}\right)+\ldots \right] \end{array}$$

Here, $p_{rgh}$ is the modified pressure $p_{rgh}=p-\rho_{f}\left(\vec{g\;}\cdot \vec{h\;}\right)$, with $\vec{h\;}$ as the height of the fluid. The scalar $\beta_{Si}$ linearly interpolates the difference in the *i*th solute concentration—compared to a reference concentration $S_{iR}$—to a difference in fluid density.

The cells were assumed to be non-motile strains for simplicity. Non-motile strains show an increased response to microgravity effects compared to motile strains, which may be due to their inability to migrate in their fluid environment [17,19,44]. Moreover, for simplicity, Brownian motion was not incorporated because the phenomenon is independent of the microgravity environment and can be approximated as negligible in solid biofilms and adherent cultures [45]; however, Brownian motion can be incorporated in future implementations of the model since it is a built-in LAMMPS feature. Brownian motion-based diffusion is about one thousand times smaller than solute diffusion [46] and was accounted for long-term in the creation of initial cell Gaussian distributions at the start of the simulations.

### 2.5. Chemical Models

We assumed the chemical concentrations within the cell to be uniform at the chemical timestep, making the following Equation (6) suitable for mass transport into the cell. The rate of substrate intake is dependent on the cell surface area $A_{C}$, the mass transfer coefficient $h_{S}$, and the substrate concentration difference from the fluid parcel outside of the cell to the inside of the cell.
(5)$$\frac{dS_{C}}{dt}=A_{C}h_{S}\;\left(S-S_{C}\right)$$

The mass transfer coefficient $h_{S}$ is related to the local fluid parcel Sherwood number (6). A Sherwood number correlation for CFD-DEM flows that were empirically determined was provided by Deen et al. [47].
(6)hSDM/2r=Sh=7−10ε+5ε21+0.17ReP15 Sc13+1.33−2.31ε+1.16ε2ReP710 Sc13 ;    0 ≤ReP ≤100

ε is the fluid parcel volume fraction, Sc is the Schmidt number, and the particle Reynolds number ReP is between 0 and 100.

We assumed that the growth was directly dependent on one substrate only. For *E. coli*, the rate-limiting substrate was methionine, and for *S. enterica,* acetate. Because we assumed their growth was rate-limited as a phenomenon of mass transfer, and not of internal cell metabolism, we assumed zero-order kinetics for each cell to convert the substrate into a product once the substrate was transported into the cell [48]. Using biomass yield ratios (Appendix A), quantities of substrate were scaled to cell biomass equivalents and then nondimensionalized using the standard cell mass. For *S. enterica*:(7)$$\left(1\;cell\;equivalent\right)\;Acetate\overset{yields}{\to }(Y_{MA}\;cell\;equivalents)\;Methionine\;\;\;\;k=\mu_{\max,S.\;enterica\;}$$
compared to *E. coli*:(8)$$\begin{array}{c} \left(1\;cell\;equivalent\right)\;Methionine+\left(1\;cell\;equivalent\right)\;Lactose\overset{yields}{\to }(Y_{AM}\;cell\;equivalents)\;Acetate\;\;\\ k=\mu_{\max,E.\;} \end{array}$$

### 2.6. Biological Models

We set up all simulations to occur during the exponential growth phase, and thus all cells were actively growing during the simulation. The Monod kinetic model for single-cell growth, where $\left[S\right]$ is the media bulk substrate concentration, is commonly used in agent-based and metabolic cell models [49].
(9)$$\mu =\mu_{\max\;}\frac{\left[S\right]}{K_{S}+\left[S\right]}$$

CAMDLES uses an altered Monod kinetic model that directly ties Monod kinetics to rate-limiting mass transfer [48]. Here (10), $\left[S_{f}\right]$ and $\;\left[S_{C}\right]$ are the concentrations of the local fluid parcel surrounding the cell and within the cell, respectively. Thus, $K_{S}$ differs from that seen in (9) and those seen in empirical growth data.
(10)$$\mu =\mu_{\max\;}\frac{\left(\left[S_{f}\right]-\left[S_{C}\right]\right)}{K_{S}+\left(\left[S_{f}\right]-\left[S_{C}\right]\right)}$$

In Equation (11), the Monod half-velocity constant $K_{S}$ is related to the heat transfer coefficient, the biomass density, the diameter of a spherical cell, the maximal growth rate, the biomass–substrate yield ratio, and the mass transfer coefficient.
(11)$$K_{S}=\frac{1}{h_{S}}\;\frac{\rho_{c}d_{c}\mu_{\max\;}}{6Y_{XS}}$$

At the simulation timescale of a few minutes only, we expected no cell division to occur. Thus, we refer to the growth rate as the individual cell increase in mass. Because we focused on relative growth rates in comparing RWV and microgravity environments, we assumed the metabolism to cause anabolic growth only, with no maintenance or decay terms like those found in NUFEB. This assumption is valid for the mass transfer-based Monod kinetic model (10) [50].

## 3. Results

Experiment inputs for all simulations described in this manuscript are given in Appendix A and input files are available at the Zenodo data repository (doi: 10.5281/zenodo.6369617). In all simulations, we found that *E. coli* and *S. enterica* growth were directly coupled. As one grew, so did the other at a rate based on the ratio of their cell yield ratios: $\frac{Y_{AM}}{Y_{MA}}$. To simplify the presentation, we depict only one species or metabolite at a time in figures, choosing to represent the species with more growth. Mean growth rates across a species are presented only when the simulation ran to steady-state growth. We report relative growth rates as a percentage of the maximum growth rate: *μ_max_* = 1.82 × 10^−5^ s^−1^ for *E. coli* and *μ_max_* = 9.09 × 10^−6^ s^−1^ for *S. enterica*.

### 3.1. Microgravity

For the first set of CAMDLES simulations, we tested the hypothesis that co-localization is necessary to enable high growth rates in the simplest environment: no gravity force. We sought to characterize how, in this condition, the spatial arrangement of the microbial community members influenced their ability to exchange metabolites and grow.

#### 3.1.1. Colocalization or Starting Metabolites Could Initiate Growth after Inoculation

Laboratory experiments are often initiated with a low population density. We tested an inoculation cell count of one cell per species in the simulation (equivalent to ~10^5^ cells/mL in a larger volume) and assessed the dependence of growth on spatial organization. In a typical experiment inoculated at low density, we can expect some *E. coli* and *S. enterica* cells to colocalize via random chance. We compared three levels of spatial organization with the placement of two cells ranging from as close as possible (1 μm) to as far away as possible (218 μm). We observed that at this cell count, the steady-state growth rate depended on the spatial organization under normal conditions (*μ* = 1.2 × 10^−6^ s^−1^, 4.9% of maximum growth rate when close together, compared to 0.8% when far apart). However, we also tested the effect of starting cells with high intracellular levels of the opposite species’ metabolite (0.01 cell equivalents of acetate or methionine, as compared to a nominal 0.00001 cell equivalents), and found that in this situation, the dependence of growth on spatial organization was mostly lost; cells grew at *μ* = 2.9 × 10^−6^ s^−1^ when closest together and at *μ* = 2.7 × 10^−6^ s^−1^ when farthest apart.

#### 3.1.2. Early-Phase Growth Rates Increased with Time and Population Density

In the early, low-density growth phase (100–283 cells in the simulation, ~10^6^ cells/mL), we assumed that cells randomly divide and distribute throughout liquid media. Thus, we set diffusion rates in the early growth phase to be that of the liquid culture. We tested a counterexample with both 100-cell and 283-cell spherical-Gaussian-packed colonies and found their steady growth rate to be hindered by ~10% compared to a randomly distributed simulation placement (Appendix A), likely due to spatial interference for diffusion or Monod competition for metabolites. In contrast, at higher population counts (2000 cells in the simulation, ~10^8^ cells/mL), the exchange of metabolites was most rapid in a compact clump of cells (Figure 4). Overall, we found that for the configuration with random spatial distributions, larger populations reached higher growth rates but with diminishing returns as populations approach densities on the order of ~10^7^ cells/mL, and never reached the growth rates achieved by densely packed cells (Figure 4).

At low cell counts, growth did not reach a steady-state rate, and it may likely never be reached within simulation timescales. Thus, we interpreted non-steady growth as a marker for a growth regime: the only way for average culture growth rates to increase in the early phase is for cells to divide and spread out.

#### 3.1.3. Growth Rate Was Limited by Diffusion Rate with Increasing Colony Size

In a cross-feeding community, the growth rate is non-linearly dependent on the total number and density of cells: as described above, there are some regimes in which more cells lead to faster growth, but in colonies of many densely packed cells, the biomass and extracellular matrix (ECM) may limit the diffusion rate of metabolites. In an experiment, colonies may densely or loosely pack depending on the gravity platform: microgravity alone may result in the formation of loose cell aggregates or colonies suspended in a liquid medium simply because there is insufficient mixing to separate the cells as they replicate.

To investigate these different colony-like scenarios, we compared growth rates in two different diffusion situations: solid biofilms and loose liquid aggregates. We kept all other product yield and cell ratio parameters at baseline (1.86 *E. coli*:1 *S. enterica*). We sought to find the population size at which a microbial community growth rate was optimal by measuring growth rates in colonies with different numbers of cells using diffusion parameters seen in liquid culture compared to those seen in a solid biofilm with an ECM (Appendix A).

We found that if the multispecies community formed an ECM (solid colony case), the growth rate was limited by diffusion as colonies increased in size: it increased with increasing colony size up to about 500 cells and then plateaued (Figure 5). Thus, we can expect that if colonies form solid aggregates, we will see them most commonly at this size in microgravity and RWVs. However, if the multispecies community does not form an ECM (liquid colony case), but rather is held together only by the quiescent fluid environment of microgravity, product yield ratios, rather than diffusion, become the limiting growth factor (see Section 3.2.3 below). Thus, for small population sizes, the growth rate in low diffusivity biofilm-like colonies is higher than in high-diffusivity liquid colonies, but the effect is reversed for large populations. In our simulations, the solid–liquid transition occurred with a colony size of about 500 cells.

#### 3.1.4. Spatial Dependence Maintained Species Ratios and High Populations

The spatial structure becomes more significant with higher populations (~10^8^ cells/mL) and more challenging growth conditions. We made growth harder by increasing the *E. coli*:*S. enterica* population ratio, achieving a methionine deficiency and thus having the overall effect of amplifying the dependence on the spatial structure. We visualized the concentration field for acetate after three minutes of simulation (Figure 6). The multispecies colony exhibited growth, while the cells that were random-uniformly distributed grew at a much lower rate.

### 3.2. Rotating Wall Vessel

#### 3.2.1. Rotating Wall Vessel Increased the Metabolite Utilization Rate

To evaluate the effect of RWV-like rotation on metabolite exchange and cell growth rates, we ran simulations using the same parameters as previously tested for microgravity with 10^8^ cells/mL (Section 3.1.2) but implemented gravity and a rotation rate of 10 R.P.M. at a distance of 1 cm from the center of rotation. We found that when the cells were randomly distributed, hydrodynamics only slightly affected RWV growth rates (29% of *μ_max_*) relative to true microgravity (27% of *μ_max_*). The slight increase was caused by metabolite convective transfer coefficients increasing to 1.17 × 10^−3^ m/s from 1.06 × 10^−3^ m/s. However, when the cells were clumped, complicated hydrodynamic effects modulated the growth dynamics and resulted in very different growth between true microgravity and an RWV. We tested a tightly packed colony in RWV conditions (Figure 7) and found that the direction of the acetate gradient was reversed compared to microgravity. In microgravity, the radial diffusion gradient from the colony was negative, which indicated that the colony was producing excess metabolites. In comparison, the gradient from a colony in RWV conditions was positive, which we interpreted to show that growth was not sustainable.

In an RWV, the growth rates decreased with time (Figure 8b); in this condition there was higher substrate utilization at initialization, but overall, lower growth rates in a steady state.

#### 3.2.2. Separated Colonies Grew Slowly in Both the RWV and Microgravity

For this obligately cross-feeding consortium, separated single-species colonies—which may form from adherent cells—are unfavorable for growth compared to multispecies colonies. Because the spatial distribution of the two species in this case was intermediate compared to randomly distributed or densely packed cases, it could provide insight into the dominating growth-limiting mechanism (diffusion vs. convection vs. production), while controlling for population density. We tested the interaction between two single-species colonies in a regime with our baseline intermediate level of diffusion (Appendix A) and like the randomly distributed case (Section 3.1.4), found very little difference between community growth rates in microgravity and RWV (Figure 8). This is unlike other cases, such as a single multispecies colony and a 1 g gravity biofilm (Figure 8b). In the separate-species case, the diffusion between colonies was the limiting factor; therefore, the complex dynamics of rotation did not influence the cross-feeding community growth rates.

#### 3.2.3. Product Yield Affected Growth Differently in RWV versus Microgravity

During a late-stage culture, multispecies colonies allow for the most rapid growth. We returned to simulating a multispecies colony and controlled for the population to measure the growth rate modulation by the balance of diffusion, convection, and production. We probed this relationship by testing the effect of the RWV hydrodynamic environment in simulations with various product yield rates *λ_S_* (*S. enterica*) and *λ_E_* (*E. coli*)*.* The product yield parameter *λ_x_* expresses the amount, in cell equivalents, of resource *x* produced during the growth of the producing species, and the ratio between *λ_S_* and *λ_E_* influences both the species ratio and the growth rate of the consortium [29]. We found that the advantage of an RWV over microgravity in terms of growth rate decreased as the product yields increased (Figure 9). However, when using baseline *λ_S_* and *λ_E_* values, the overall effect of the RWV hydrodynamic environment was unclear, as it depended on diffusion parameters. We tested two sets of diffusion parameters—those found in biofilms/ECM (Figure 9a) and those found in liquid media (Figure 9b)—and found that the RWV overall increased the growth rates when we assumed low diffusion but decreased growth rates when we assumed high diffusion.

### 3.3. In 1 g Gravity

#### 3.3.1. Gravity Induced the Spatial Structure

The reduced sedimentation exhibited by RWV and microgravity may itself change the spatial structure of colonies. To compare the 1 g gravity effect of sedimentation, we placed either a 2D circular biofilm (growth trajectory depicted in Figure 8b) or a spherical biofilm (Figure 10) on the bottom floor of the simulation box. We found that the two conformations had different growth rates, indicating that reduced growth in 1 g gravity could be caused by the geometry of the colony or by its localization on the floor. When cells sediment to the floor, diffusion was restricted upward, where the overall impact depended on the colony geometry.

#### 3.3.2. Natural Convective Flow Was Negligible

Our simulations also showed that in 1 g, natural convection in a biofilm could result in up to a 20% increased growth rate relative to microgravity. Yet, when the diffusivity was increased to that of liquid media, the growth rate only increased by 3% as the metabolite convective plume diffused. However, it is worth acknowledging that the effect of convection in 1 g (and thus RWV) in our simulation conditions was negligible when compared to particle Brownian convection (Brownian diffusivity ~10^−9^). When multiplied by a maximum simulation Péclet number of ~0.05, the realistic convective effect was ~0.1–1%. A natural convective flow is displayed (Figure 10) in 1 g gravity.

## 4. Discussion

In this study, we used CFD-DEM modeling to investigate the relationship between the physical environment and the ecological interactions in a microbial community. We chose to simulate a community for which similar research questions were addressed using a very different form of modeling, namely, dynamic flux balance analysis and metabolite diffusion on a lattice, using the COMETS platform [28]. Several contrasts are worth noting. CAMDLES assumes that the internal metabolism time is negligible compared to the time for metabolites to transfer throughout liquid media, and thus we effectively linearized the Monod kinetics equation. Importantly, this approximation is more accurate the lower the diffusivity of metabolites. When simulating low diffusivity, we observed growth rates around 10–20% of *μ_max_*, but with high diffusivity, we observed up to 35% of *μ_max_*. Accordingly, the general growth results of this study agreed with the ~35% relative growth rate seen in vitro in 1 g gravity [30].

Diffusivity affects the extent to which community growth is dependent on the spatial distribution of cells. Unfortunately, in CAMDLES, we could not model the effects of non-cellular solids, such as an extracellular matrix (ECM), on diffusivity. In vitro and in silico effective diffusivities decrease in solid biofilms [51,52], limiting mass transfer overall. Baseline diffusivities for each metabolite were set as constant in CAMDLES; we were able to change these values to explore differences between biofilm-like and non-biofilm situations, but such changes were global and, therefore, unable to reflect far-field effects outside of colonies within a larger liquid culture vessel. Moreover, the metabolic state of the cells within the biofilm was not resolved, as we assumed all cells to be actively growing.

The specific goal of this work was to compare the conditions of artificial microgravity and spaceflight microgravity and assess which factors led to the most similar results between the two conditions. When solid biofilms form in RWVs, bigger spherical aggregates and more growth are favored relative to those in microgravity. In addition, the ECM effect on diffusion may also increase growth rates compared to those in microgravity. However, if colonies are loose, liquid aggregates, RWVs may reduce the spatial structure of non-motile cells. In summary, RWVs generally amplify the spatial importance of this multispecies microbial community, leading to different aggregation sizes and population ratios. This may change, however, depending on the product yield rates and population proportions of the species.

A specific insight that CAMDLES provides is its ability to predict when optimal community growth conditions vary between RWV and microgravity. We found that in the low-population stage of metabolite exchange, planktonic growth was favorable. In our simulations, low-population growth rates increased over time. Later, as cross-feeding metabolites accumulated, planktonic growth rates approached a maximum level. Higher growth rates then only occurred when cells aggregated into either a densely packed biofilm or a loose spherical aggregate. These results parallel observations showing that bacteria can exhibit a planktonic/biofilm transition at a critical point [52], and hint that future CAMDLES studies could offer insight into growth strategies underlying multispecies biofilm formation. In 1 g and RWV cases, forces will likely break apart spherical aggregates and, thus, we would expect to observe mostly spherical, solid-biofilm formation, but organisms might evolve different strategies in microgravity.

More work is necessary to determine whether our simulation domain size was large enough to capture the diffusion dynamics of metabolites; although it is large enough to contain a single colony, it did not consider all possible interactions with multiple colonies, except for the simplest, uniform interaction from choosing periodic boundary conditions. Additionally, since we had to prescribe the population size, aggregation, and colocalization of species during the start of the simulation, we were only able to compare relative growth rates in specific conditions. While this gave us the advantage of fine-tuning the experiments to compare diverse culture conditions, it limited the conclusions we could make about long-term growth dynamics. Because of the difference in biological and hydrodynamic timescales, CAMDLES simulations did not run long enough to encounter cell division. However, the timespan was large enough that, except for low population levels, steady-state growth (cell mass accumulation) rates were achieved.

At the evolutionary scale, one predictor of the potential size of biofilms or cell aggregates could be the point at which the optimal spatial arrangement transitions from planktonic to aggregated according to different diffusivities. At a certain point, an ECM reduces diffusion and limits growth rates, and thus the colony size, of solid-spherical biofilms. In CAMDLES, this occurs at a colony size of around 500–1000 cells. Thus, we expect that if colonies form solid aggregates, we will see them most commonly at this size in microgravity and RWVs.

We underscore that these results hold only for multi-species aggregates and not for single-species aggregates. In contrast to agar plates, where single-species colonies can support each other at a distance, multi-species colonies are necessary for metabolite exchange in liquid media in all three conditions tested (RWV, microgravity, 1 g). One potential explanation concerned two disadvantages: single-species colonies exhibit both the poor diffusive transport that cells in a planktonic culture experience and the intra-colony competition for metabolites that cells in dense communities experience. Yet, CAMDLES simulations do not resolve the long-term spatiotemporal effects of an RWV and 1 g for these scenarios. The long-term effect is simplest in a quiescent environment of microgravity, but interactions between multiple colonies would be more complicated in an RWV sedimenting environment.

Overall, we observed that the differences between microgravity and artificial microgravity originated primarily from increased mass transfer caused by RWV convective flows and sedimentation. Even though convection fluid currents may be negligible in 1 g and RWVs, the increased mass transfer coefficient of cells $h_{S}$ resulted in a 9% reduction in the Monod half-saturation constant $K_{S}$. The spatiotemporal dynamics and substrate utilization of the colony itself were changed by the $K_{S}$ change in both species. However, these mass transfer coefficient increases were minor in comparison to those that would result from Brownian motion if it were implemented, which is significantly more rapid than RWV sedimentation [53]. On the other hand, it is limited in the case of solid biofilms [45]; therefore, we can assert that solid biofilms are more accurately simulated in CAMDLES. Generally, if Brownian motion and cell motility were included in the simulation, we would likely observe a reduction in the relative difference between microgravity and RWV growth rates, and an overall reduction in growth rates due to long-term biomass dispersion throughout the simulation domain.

## 5. Conclusions

CAMDLES is designed to be easily modifiable and extendable. It retains the modularity and support that is featured in the open-source software LAMMPS and OpenFOAM. Further functionality with CAMDLES includes exploring the effect of motility, Brownian motion, cell–cell adhesion, or multisphere support for rod-shaped bacteria on microbial cell dynamics. These general features are already included in LIGGGHTS^®^/LAMMPS. Additionally, CAMDLES retains the functionality of ParScale, which can model transport and chemical reactions within a spherical particle. This multiscale simulation functionality provides new avenues for microbial agent-based modeling in RWVs.

The greatest limitations of CAMDLES stem from size and timespan constraints. The simultaneous fluid, particle, and chemical models incorporated in CAMDLES limit the timescale of simulation to only a few minutes on a personal computer. Performance has yet to be analyzed on high-performance computing clusters. However, CAMDLES retains the highly efficient parallelization found in LIGGGHTS^®^ and CFDEM^®^Coupling. Added features, such as Brownian motion, bacterial motility, and more complex metabolic cell models, would also add computational overhead. In the future, computational cost can be relieved by developing coarse-grained models for CAMDLES. In conjunction with agent-based models, employing a computationally efficient, coarse-grained modeling approach can provide detailed conclusions regarding the multigenerational evolution and population dynamics of microbial communities.

Based on our simulations, we found artificial microgravity to have an amplifying effect on mass transfer and metabolite uptake compared to microgravity and a diminishing effect on sedimentation compared to 1 g gravity. Although not tested in this study, the size and rotation rate of the RWV balance these two factors, and in future work, CAMDLES could be used to explore these experimental variables. Ultimately, the RWV effect on population growth rate is unclear and specific to the microbial community of study. Thus, our package has the potential to serve as a basis to demystify the inconsistencies seen in microbial communities between RWVs and microgravity. This first implementation of CAMDLES serves to demonstrate the importance of a microbial cell’s immediate physical environment on population- and community-level behaviors, especially in the space environment, and in experiments meant to simulate the space environment. CAMDLES represents the type of computational tool capable of providing biologists with unprecedented insight into those interacting physical, chemical, and biological processes.

## Figures and Tables

**Figure 1 life-12-00660-f001:**
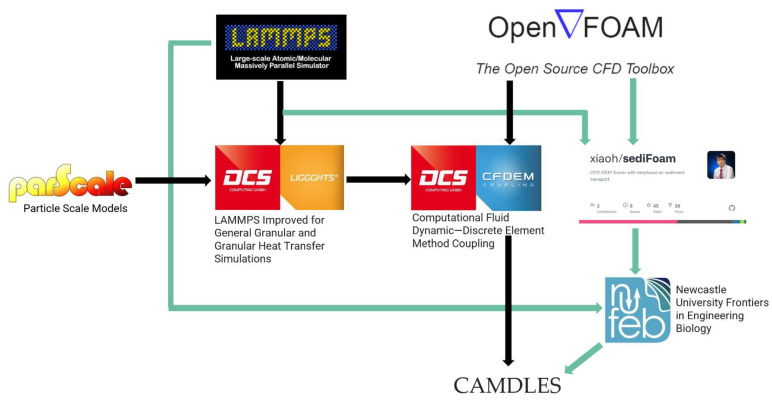
Software technology tree. CAMDLES relates to parent software via black arrows and related software via green arrows.

**Figure 2 life-12-00660-f002:**
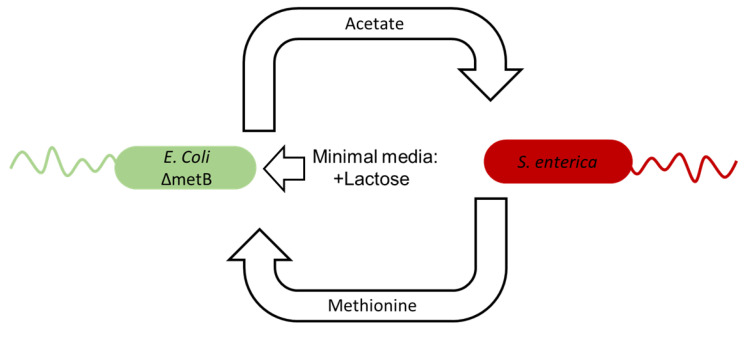
Model microbial community simulated in this study. Adapted with permission from [28]; published by *Cell Reports*, 2014. When grown in minimal media with lactose, *Escherichia coli* K12 with a *metB* gene knockout (*E. coli* Δ*metB*) provides acetate as the carbon growth substrate necessary for a methionine-secreting mutant of *Salmonella enterica* ser. Typhimurium (*S. enterica*) to grow. In exchange, the *S. enterica* provides the methionine necessary for the *E. coli* to grow.

**Figure 3 life-12-00660-f003:**
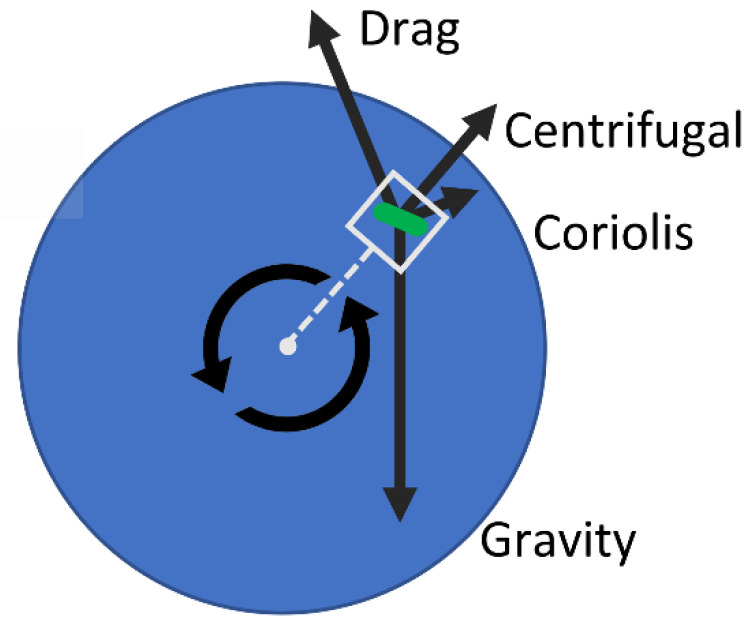
Free body diagram of a particle immersed in an RWV. Acceleration terms are described in Equation (3). The simulation domain is a rectangular box in a rotating reference frame.

**Figure 4 life-12-00660-f004:**
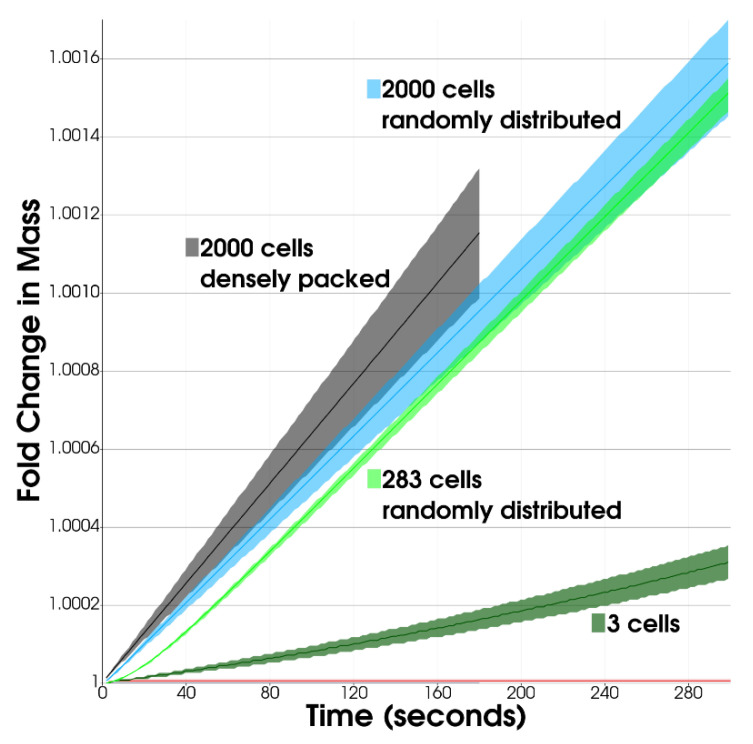
Average *E. coli* Fold Change in Mass vs. Colony Population in Microgravity. Solid lines represent statistical means, while transparent wedges represent inner quartiles. Either densely packed spherical or randomly distributed colonies were generated at various sizes. Only one colony was generated per simulation. Note: as the population density increased, the standard deviation of growth rates increased, which was likely caused by local spatial heterogeneity within the simulation domain, increasing the likelihood of resource competition.

**Figure 5 life-12-00660-f005:**
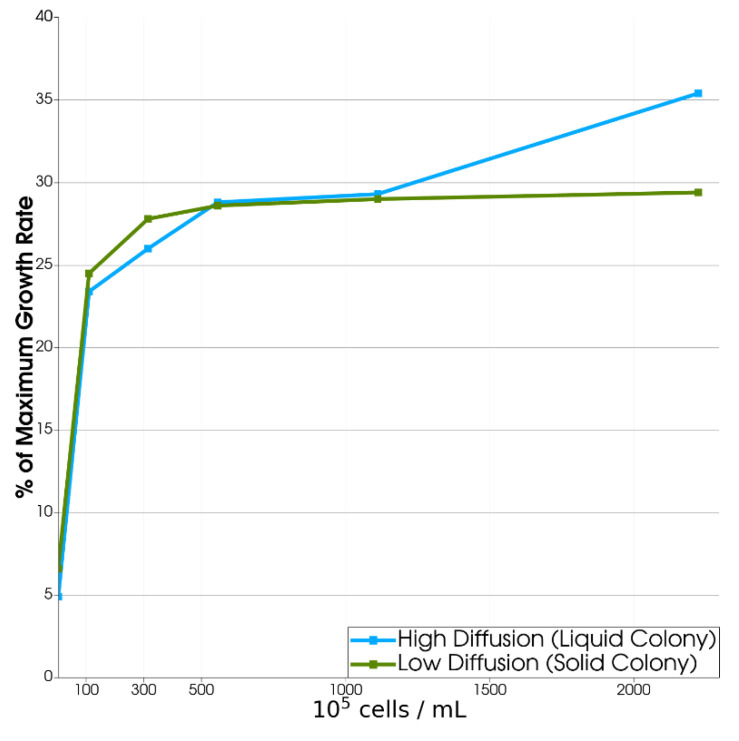
*E. coli* Percent Maximum Growth Rate vs. Colony Population in Microgravity. Densely packed multispecies spherical colonies were generated at various sizes. Only one colony was generated per simulation. In solid colonies, self-insulation limited the growth rate beyond a colony of 500 cells. The effect was reduced in liquid colonies.

**Figure 6 life-12-00660-f006:**
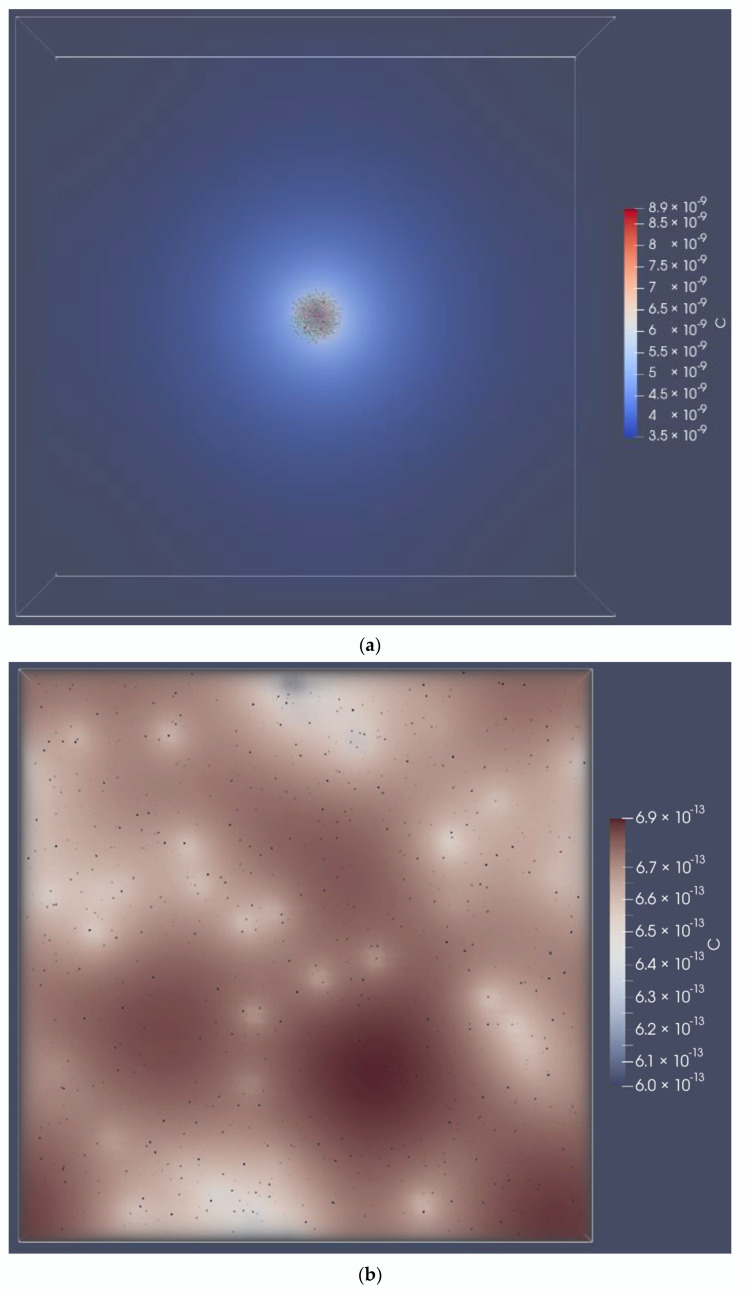
Acetate diffusion. The symbol C represents the acetate concentration in units of cell equivalents/μm^3^; note that the color scale is different between the two panels. Compared to other simulations, the *E. coli*:*S. enterica* population ratio was increased to a 1956:44 count. Diffusion parameters were set to be equal in (**a**,**b**) at a nominal 5 × 10^−6^ cm^2^/s. (**a**) Two-species spherical colony; (**b**) random uniformly distributed cells.

**Figure 7 life-12-00660-f007:**
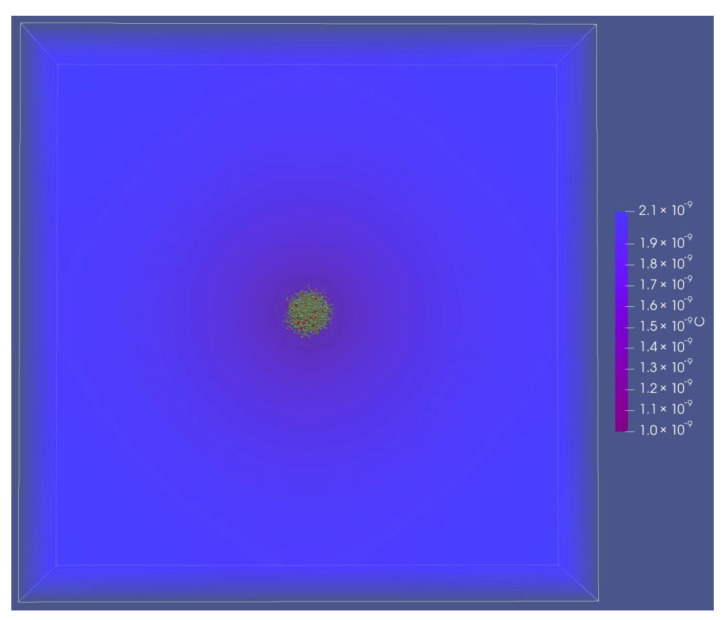
Acetate diffusion in RWV conditions. The color scale (symbol C) represents acetate concentration in units of cell equivalents/μm^3^. Note that the direction of the diffusion gradient was reversed in this case compared to that found in microgravity (Figure 6a) and that the acetate concentrations decreased. The reversed gradient indicated that the growth rates were unsustainable.

**Figure 8 life-12-00660-f008:**
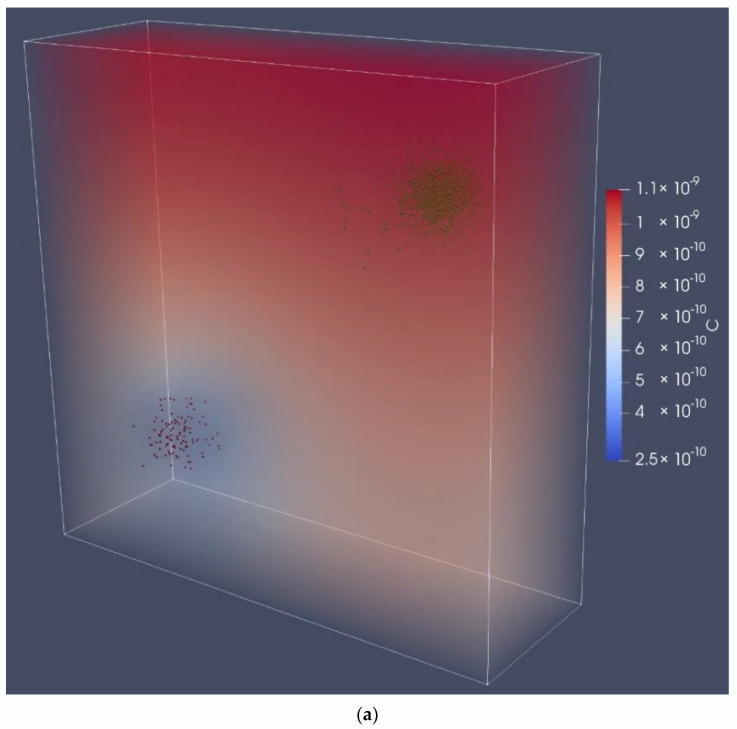
Single-species Colonies Distanced Far apart. The *E. coli*:*S. enterica* population ratio was adjusted to optimal conditions (1895 count:105 count). (**a**) Coculture colonies of *E. coli* (green) and *S. enterica* (red). The symbol C represents acetate concentration in units of cell equivalents/μm^3^. Colonies were generated as spherical Gaussian distributions placed as far away as possible within the computational domain. The microgravity case is displayed, but the RWV case was near-identical. (**b**) Collection of *S. enterica* growth over time in tightly packed colonies. The centerline plots the average cell mass. The dark shaded region shows the inner quartiles, and the light shaded region shows the outer quartiles. The growth trajectories for the cases depicted in Figure 6a and Figure 7 are graphed here.

**Figure 9 life-12-00660-f009:**
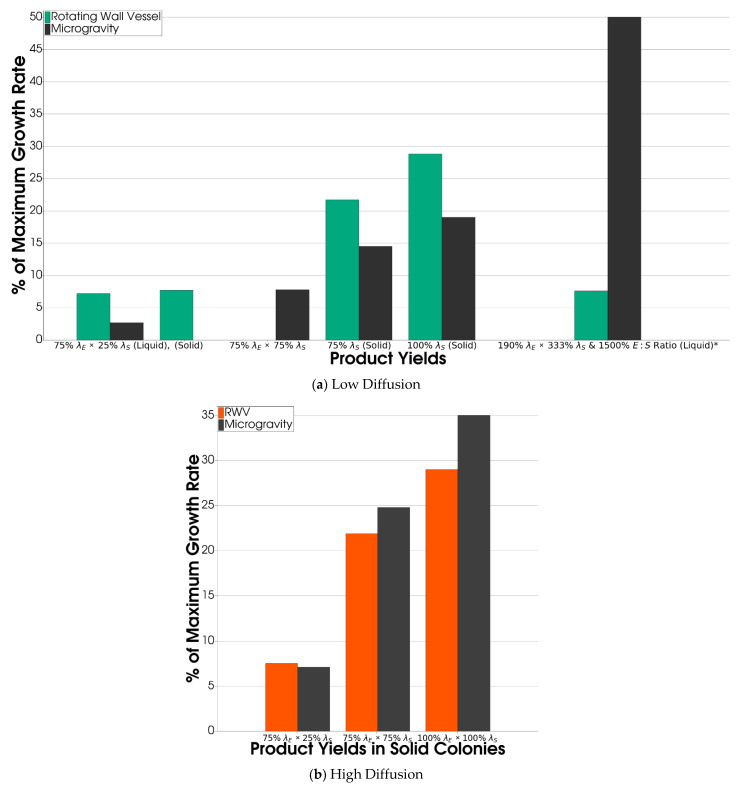
*E. coli* Percent Maximum Growth Rate with Various Product Yield Parameters *λ*. If not specified, *λ* was set to 100% as the default. Densely packed multispecies spherical colonies were generated with a colony size of 2000 cells. Only one colony was generated per simulation. In an RWV, gravity caused the colonies to move as either a rigid unit (regarded as a solid) or each cell moved individually (regarded as liquid). This was not important to the growth rates and was only tested in: (**a**) an RWV vs. microgravity with low diffusion parameters like those in biofilms. * *S. Enterica* growth is displayed as it was the faster-growing species, and the diffusion was at an intermediate level. (**b**) An RWV vs. microgravity with high diffusion parameters like those in liquid media.

**Figure 10 life-12-00660-f010:**
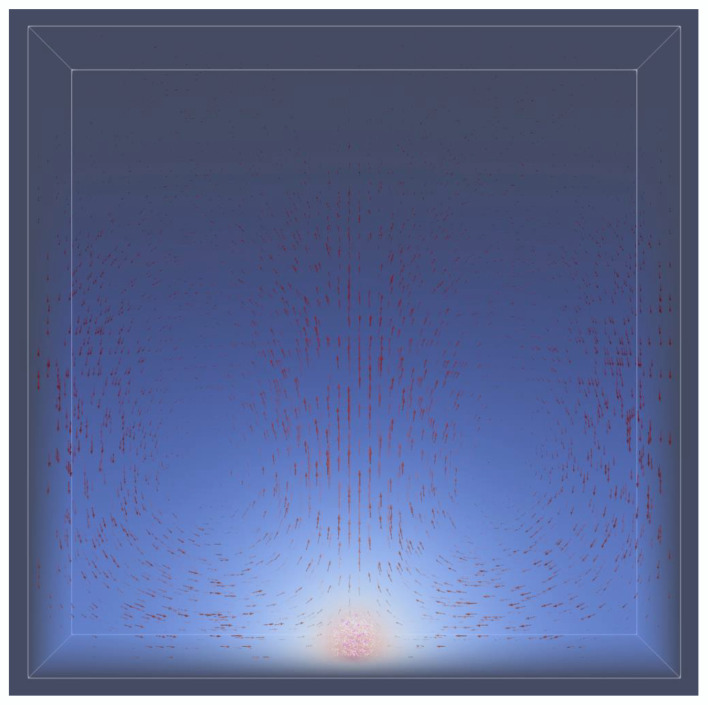
Natural Convective Flow. A spherical biofilm placed on the simulation floor exhibited solute convective flows in a 1 g gravity condition. Red velocity arrows, in units of s^−1^, are scaled up by 1000×. The concentration field of acetate is depicted from high to low as red to blue. Fluid simulation boundaries are periodic except on the floor and ceiling of the box.

**Table 1 life-12-00660-t001:** Parent and related software. Parent software includes CAMDLES dependencies or packages derived from dependencies and related software includes packages that indirectly inspire CAMDLES development.

Name	Publication Year	Description and Relevant Features	References
**Parent Software**			
LAMMPS (Large-Scale Atomic/Molecular Massively Parallel Simulator)	1995	Classical molecular dynamics simulator Brownian motion Gravity	[34]
LIGGGHTS^®^ (LAMMPS Improved for General Granular and Granular Heat Transfer Simulations)	2012	Enhanced support for larger granular particles Heat and mass transfer	[33]
OpenFOAM (Field Operation and Manipulation)	1998	Computational fluid dynamics package Single reference frame Heat convection buoyant Boussinesq approximation	[35]
CFDEM^®^Coupling	2012	Couples LIGGGHTS^®^ and OpenFOAM Scalar transport Robust coupling models [36]	[32]
**Related Software**			
SediFOAM	2017	Alternative solid–fluid coupling approach of LAMMPS and OpenFOAM	[37]
NUFEB (Newcastle University Frontiers in Engineering Biology)	2019	Agent-based biological extension of SediFOAM Energy-based and diffusion-based Monod Kinetics Biological growth	[31]
ParScale (Particle Scale Models)	2017	Intra-particle transport models coupled to LIGGGHTS^®^ and CFDEM^®^Coupling Single chemical reaction	[38]
COMETS (Computation of Microbial Ecosystems in Time and Space)	2021	Population-based flux balance analysis with spatial growth and diffusion Evolution Multiple species/genotypes	[39]

## Data Availability

The data presented in this study are available within the article and Appendix A, and/or can be reproduced using the code in File S1 and experiment input files available at the Zenodo data repository (doi:10.5281/zenodo.6369617).

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
