# Peer review of "CAMDLES: CFD-DEM Simulation of Microbial Communities in Spaceflight and Artificial Microgravity"

_life, 2022, doi:10.3390/life12050660_

Round 1

Reviewer 1 Report

The paper is well introduced. Title reflects complexity and main objectives of the paper.

1. General comments: Strenghts and limitations: Strenghts: good philosophy and methods, 
systematic research, innovative and applicable research. Limitations: lack of hypothesis, results exploration. 
about multispecies community forms . Please, take a look at: an English  revision (some mistakes).

The  lengthy conclusion section needs to be improved. Usually, this section should be concise and succinct so that readers could have a final chance to review the most important points obtained from the article. However, the current one is a bit long and I believe it could
be compacted into one or two paragraph but the main points could still be kept.

Author Response

Thank you for your comments!

Both authors, as well as one additional colleague, have read through the manuscript again to correct any typographical errors.

To address the comment on the lengthy conclusion section, we have reformatted and reworded the Discussion session, and separated some of the text into a brief Conclusion that summarizes our findings. The Life Instructions for Authors note that a Conclusion section is optional but may be included in papers with long or complex Discussion sections.

Reviewer 2 Report

Journal name: Life (MDPI Publications)

Manuscript ID: life-1665717

Title: CAMDLES: CFD-DEM Simulation of Microbial Communities in Spaceflight and Artificial Microgravity

Authors: Rocky An and Jessica Audrey Lee.

The manuscript is well written and, the study has significant importance in the field of microbial technology and metabolic engineering. The manuscript can be accepted in the present form. In addition, it requires some minor corrections that to be corrected before publication in the prestigious Journal “Life” (MDPI)

Minor Comments#

  1. Keywords is too many, author could provide precise key points!
  2. Please make uniform citation in the text (it looks some ref. fold and unfold)
  3. Table 1: authors provided details about parent and related software’s that used for CAMDLES development. Also, authors can add some more detail about software sources/ internet web ID, etc.

Is this (software) can use/access general researchers?

  1. I appreciate the authors that figure 2 adapted and modified well. Though, author can elaborate in detail the figure caption, what is ∆metB, bacterial strains ( coli, S. enterica) abbreviation….!

Author Response

Thank you very much for your positive feedback and for your specific suggestions. We have copy each of the minor comments here and address them individually:

1. Keywords is too many, author could provide precise key points!

We have removed three of the keywords, leaving seven.

2. Please make uniform citation in the text (it looks some ref. fold and unfold)

We have reformatted to ensure that all citations are not bold and not italicized.

3. Table 1: authors provided details about parent and related software’s that used for CAMDLES development. Also, authors can add some more detail about software sources/ internet web ID, etc. Is this (software) can use/access general researchers?

We have edited Table 1 to include the requested information, and added urls for accessing the referenced software in the bibliography.

4. I appreciate the authors that figure 2 adapted and modified well. Though, author can elaborate in detail the figure caption, what is ∆metB, bacterial strains (coli, S. enterica) abbreviation….!

We have elaborated in the caption to Figure 2 as requested.